# Long Intergenic Noncoding RNA *OIN1* Promotes Ovarian Cancer Growth by Modulating Apoptosis-Related Gene Expression

**DOI:** 10.3390/ijms222011242

**Published:** 2021-10-18

**Authors:** Toshihiko Takeiwa, Yuichi Mitobe, Kazuhiro Ikeda, Kosei Hasegawa, Kuniko Horie, Satoshi Inoue

**Affiliations:** 1Division of Systems Medicine & Gene Therapy, Saitama Medical University, Hidaka, Saitama 350-1241, Japan; ttakeiwa@tmig.or.jp (T.T.); ymitobe31@gmail.com (Y.M.); ikeda@saitama-med.ac.jp (K.I.); 2Department of Systems Aging Science and Medicine, Tokyo Metropolitan Institute of Gerontology, Itabashi-ku, Tokyo 173-0015, Japan; 3Department of Gynecologic Oncology, Saitama Medical University International Medical Center, Hidaka, Saitama 350-1298, Japan; koseih@saitama-med.ac.jp

**Keywords:** long intergenic noncoding RNA (lincRNA), ovarian cancer, RNA sequencing, small interfering RNA (siRNA), xenograft

## Abstract

Patients with advanced ovarian cancer usually exhibit high mortality rates, thus more efficient therapeutic strategies are expected to be developed. Recent transcriptomic studies revealed that long intergenic noncoding RNAs (lincRNAs) can be a new class of molecular targets for cancer management, because lincRNAs likely exert tissue-specific activities compared with protein-coding genes or other noncoding RNAs. We here show that an unannotated lincRNA originated from chromosome 10q21 and designated as *ovarian cancer long intergenic noncoding RNA 1* (*OIN1*), is often overexpressed in ovarian cancer tissues compared with normal ovaries as analyzed by RNA sequencing. *OIN1* silencing by specific siRNAs significantly exerted proliferation inhibition and enhanced apoptosis in ovarian cancer cells. Notably, RNA sequencing showed that *OIN1* expression was negatively correlated with the expression of apoptosis-related genes *ras association domain family member 5* (*RASSF5*) and *adenosine A1 receptor* (*ADORA1*), which were upregulated by *OIN1* knockdown in ovarian cancer cells. *OIN1*-specifc siRNA injection was effective to suppress in vivo tumor growth of ovarian cancer cells inoculated in immunodeficient mice. Taken together, *OIN1* could function as a tumor-promoting lincRNA in ovarian cancer through modulating apoptosis and will be a potential molecular target for ovarian cancer management.

## 1. Introduction

Ovarian cancer is one of the most common cancers in women [1]. Despite efforts to develop effective therapeutic strategies for ovarian cancer, the mortality rate remains the highest among gynecological cancers [2]. Because of the absence of symptoms in early stages, ~60% of ovarian cancers are diagnosed as advanced disease. The 5-year overall survival rate of ovarian cancer patients remains <50% [3]. The development of new biomarkers and therapeutic targets is expected to improve current diagnosis and treatment of ovarian cancer.

Advancements in cloning and sequencing technologies have revealed that 70–90% of mammalian genomes are transcribed into a variety of noncoding RNAs (ncRNAs) that do not encode proteins [4]. ncRNAs with more than 200 bases are classified as long noncoding RNAs (lncRNAs), while others are classified as small noncoding RNAs (sncRNAs). MicroRNAs (miRNAs) consist of a class of sncRNAs, and some miRNAs, including *miR-200c-3p,* have been indicated to be key regulators of ovarian cancer pathophysiology [5,6]. Moreover, several lncRNAs have been characterized as playing oncogenic or tumor-suppressive roles in cancers including ovarian cancer [7,8,9,10,11,12,13]. For example, *metastasis associated lung adenocarcinoma transcript 1* (*MALAT1*) is an lncRNA which is upregulated in ovarian cancer and associated with poor prognosis in ovarian cancer patients. Previous studies demonstrated that *MALAT1* promotes proliferation and suppresses apoptosis in ovarian cancer by sponging miRNAs such as *miR-211* [12]. Other reports have also indicated that *MALAT1* regulates the migration and invasion of ovarian cancer cells through controlling the expression of extracellular matrix genes and cell signaling pathways such as phosphoinositide 3-kinase (PI3K)/AKT and Wnt/β-catenin [12]. In contrast, *growth arrest-specific 5* (*GAS5*) is an lncRNA that exerts a tumor-suppressive function in ovarian cancer through sponging miRNAs such as *miR-21* [12], and by facilitating the E2F4-mediated transcriptional repression of *poly (ADP-ribose) polymerase 1* (*PARP1*) [14]. Nevertheless, the majority of lncRNAs remain to be characterized in ovarian cancer, and their functional annotation may provide useful information for potential diagnostic and therapeutic targets for the disease.

We previously performed RNA-sequencing (RNA-seq) analysis using clinical ovarian tissues and cancer specimens and identified candidate genes associated with the pathophysiology of ovarian cancer [15,16]. We also identified novel mutations in known ovarian cancer-associated genes, such as *TP53*, *breast cancer 2* (*BRCA2*), and *AT-rich interaction domain 1A* (*ARID1A*) [16]. These results indicate that RNA-seq analysis using clinical samples is useful for the characterization of ovarian cancer gene signature and the screen of new therapeutic targets. Based on the RNA-seq, we here focused on functional lncRNAs, particularly long intergenic noncoding RNAs (lincRNAs) predominantly expressed in ovarian cancer tissues. Apart from a genic lncRNA subclass that shares sequence with protein-coding transcript, the lincRNA subclass may exert tissue-specific activities, as lincRNAs derived from intergenic “gene deserts” generally show less conservation than protein-coding genes or other noncoding RNAs across species [17]. We identified an uncharacterized lincRNA originated from chromosome 10q21, whose expression was substantially higher in clinical ovarian cancer specimens compared with normal ovarian tissues. We designated the lincRNA as *ovarian cancer long intergenic noncoding RNA 1* (*OIN1*). Loss-of-function study using *OIN1*-specific siRNAs showed significant proliferation inhibition and enhanced apoptosis in A2780 and SKOV3 ovarian cancer cells. Moreover, we found that the expression of apoptosis-related genes *ras association domain family member 5* (*RASSF5*) and *adenosine A1 receptor* (*ADORA1*) could be modulated by *OIN1* in ovarian cancer. We propose that lincRNA *OIN1* could contribute to ovarian cancer progression by fine-tuning gene expression, leading to the suppression of apoptosis.

## 2. Results

### 2.1. OIN1 Is Highly Expressed in Ovarian Cancer Tissues and Cells

In the present study, we aimed to identify functional lincRNAs predominantly expressed in ovarian cancer. We screened our RNA-seq data analyzed by NONCODE database [18] obtained from clinical specimens of normal ovarian tissues (*n* = 6) and ovarian cancers (*n* = 15) [15,16]. Screening NONCODE v4 transcripts abundantly expressed in ovarian cancer compared with normal ovarian tissues by ≥10-folds at an FDR *q*-value threshold <0.05, we identified 10 particular lincRNAs (Table 1), including known oncogenic RNAs: *competing endogenous lncRNA 2 for microRNA let-7b* (*CERNA2*)/*human ovarian cancer-specific transcript 2 (HOST2)* [19], *urothelial cancer associated 1* (*UCA1*) [14,20,21,22], and *long intergenic non-protein coding RNA 958 (LINC00958)* [23]. *CERNA2*/*HOST2* was shown to promote ovarian cancer cell proliferation, partly by sponging the tumor suppressor *let-7b* [19]. In terms of *UCA1*, the elevated expression was shown to enhance cell migration, invasion and cisplatin resistance of ovarian cancer [20,21], and the lincRNA activates Hippo-Yes-associated protein (YAP) signaling in ovarian cancer [22].

Since we noticed functional ovarian cancer-related lincRNAs among the top 10 RNAs, we next characterized the most abundantly expressed lincRNA in ovarian cancer. The sequence of the top lincRNA corresponds to the *NONHSAT013448* in NONCODE database, with the mean RPKM (reads per kilobase of transcript length per million mapped reads) values as 79.2 ± 24.8 and 3.0 ± 2.6 in ovarian cancer and normal tissues, respectively (*q* = 0.006) (Figure 1A,B; Table 1). Coding potential calculator (CPC) algorithm showed that the coding potential score of *NONHSAT013448* was −0.62195, suggesting that it is a putative noncoding RNA. *NONHSAT013448* is transcribed from a gene *NONHSAG005930* at chromosome 10q21.1, whose location is ~0.78 and ~0.20 Mb apart from the neighboring protein-coding genes *protocadherin-related 15* (*PCDH15*) and *mannose binding lectin 2* (*MBL2*), respectively. We designated the lincRNA as *ovarian cancer long intergenic noncoding RNA 1,* or *OIN1*. Based on quantitative real-time PCR (qRT-PCR) analysis in ovarian cancer cell lines, we found that high expression of *OIN1* was observed in A2780 and SKOV3 cells, and moderate expression in OV90 cells (Figure 1C).

### 2.2. OIN1 Promotes Proliferation and Suppresses Apoptosis of Ovarian Cancer Cells

To examine the significance of *OIN1* in ovarian cancer, we performed knockdown experiments using *OIN1*-specific siRNAs (siOIN1 #1 and #2). *OIN1* expression was substantially decreased by these siRNAs in A2780 and SKOV3 cells as analyzed by qRT-PCR (Figure 2A). We showed that the siRNA-mediated *OIN1* knockdown significantly suppressed the proliferation of these A2780 and SKOV3 cells (Figure 2B). Conversely, the exogenous expression of *OIN1* promoted the proliferation of A2780 and SKOV3 cells (Appendix A). We next examined whether *OIN1* knockdown modulates the apoptosis of ovarian cancer cells. As analyzed by flow cytometry, *OIN1* knockdown increased annexin V-positive fractions in A2780 and SKOV3 cells (Figure 2C–F). The expression levels of an apoptosis marker, cleaved PARP1 protein, were increased in *OIN1*-silenced A2780 and SKOV3 cells (Appendix A). In addition, the mRNA expression of anti-apoptotic *B-cell lymphoma 2* (*BCL2*) was downregulated, and the ratio of the expression of proapoptotic *BCL2-associated X, apoptosis regulator* (*BAX*) to *BCL2* was increased in *OIN1*-silenced cells (Appendix A). These results suggest that *OIN1* promotes proliferation and suppresses apoptosis of ovarian cancer cells.

### 2.3. OIN1 Modulates the Expression of Apoptosis- or Cell Proliferation-Related Genes

We next screened *OIN1*-associated genes whose expression levels exhibited positive or negative correlations with *OIN1* in the RNA-seq data from ovarian cancer specimens [15,16]. We found that 323 and 312 genes had correlation coefficients of ≥0.6 and ≤−0.45 with *OIN1*, respectively, and analyzed enriched biological pathways in these identified genes using the DAVID Bioinformatics Resources 6.8 [24] (Appendix A). We noted that the apoptosis- and cell proliferation-related pathways were enriched in the genes showing negative correlation coefficient values (≤−0.45) (Appendix A). Among the selected genes, we selected 4 genes including *cyclin-dependent kinase inhibitor 1B* (*CDKN1B*)*, RASSF5, ADORA1, and RNA-binding motif protein 5* (*RBM5*), which were previously characterized as those involved in apoptosis and cell proliferation (Figure 3A). Of the 4 genes, we found that *OIN1* knockdown significantly upregulated *RASSF5* and *ADORA1* mRNA levels in both A2780 and SKOV3 cells (Figure 3A,B). In contrast, *OIN1* overexpression substantially downregulated *RASSF5* and *ADORA1* mRNA levels in A2780 and SKOV3 cells (Appendix A). Scattered plots of *RASSF5* or *ADORA1* mRNA levels versus *OIN1* levels, shown as RPKM values of our RNA-seq data of clinical ovarian cancer tissues, revealed that *OIN1* had a tendency of negative correlation with *RASSF5* (*r* = −0.49 and *p* = 0.06) and a significant negative correlation with *ADORA1* (*r* = −0.63 and *p* = 0.01) (Figure 3C). Taken together, we proposed that *OIN1* negatively modulates expression of *RASSF5* or *ADORA1*, which may contribute to the alteration of ovarian cancer cell proliferation.

### 2.4. OIN1 Silencing Suppresses In Vivo Tumor Growth of Ovarian Cancer Cells

We further examined the role of *OIN1* in in vivo tumor formation of ovarian cancer cells. We generated a A2780 cell-derived xenograft model by the subcutaneous administration of cells with Matrigel into female athymic mice, followed by the intratumoral injection of siRNAs twice a week. We showed that the *OIN1*-specific siRNA injection significantly suppressed tumor formation in the xenografted mice (Figure 4A,B and Appendix A). Notably, *OIN1* expression was downregulated (Figure 4C), whereas *RASSF5* and *ADORA1* expression was increased (Figure 4D,E) in dissected xenograft tumors injected with *OIN1*-specific siRNA. Overall, these results suggest that *OIN1* is a functional lincRNA that may contribute to ovarian cancer progression by modulating apoptosis-related gene expression.

## 3. Discussion

In the present study, we identified the novel ovarian cancer-related lincRNA *OIN1*, which is highly expressed in ovarian cancer. Functional analyses revealed that *OIN1* substantially suppresses apoptosis and promotes the proliferation of ovarian cancer cells. Moreover, candidate apoptosis- or cell proliferation-related genes *RASSF5* and *ADORA1* were identified as *OIN1* downstream genes in ovarian cancer. siRNA-mediated *OIN1* silencing significantly decreased the in vivo tumor formation of ovarian cancer cells, along with the upregulation of *RASSF5* and *ADORA1*. Although there was a difference in the experiment conditions of in vitro and in vivo *OIN1* silencing experiments, these results suggest that *OIN1* plays a crucial role in ovarian cancer progression.

Among the ovarian cancer cells used in the present study, *OIN1* was highly expressed in A2780 and SKOV3 cells, and was moderately expressed in OV90 cells. Ovarian cancer is a heterogeneous disease and is classified into multiple histological subtypes. Regarding the cells used in this study, OV90 and OVCAR3 cells are derived from high-grade serous carcinoma while ES2 cells are derived from clear cell carcinoma, and the histological subtypes from which A2780 and SKOV3 originate remain controversial [25,26]. All the cells used in this study possess wild-type *breast cancer 1/2* (*BRCA1/2*) [25,26,27]. Meanwhile, *TP53* mutations have been reported in SKOV3, OV90, OVCAR3, and ES2 cells, while A2780 cells have wild-type *TP53* [26]. Since there is no correlation between *TP53* mutation status and *OIN1* expression levels in these cells, we assume that *TP53* mutation status may not be a determinant of *OIN1* expression levels in ovarian cancer cells.

RASSF5 belongs to the ras-association domain family, which is generally known as a tumor suppressor [28]. Through the conformational change in RASSF5 upon ras association, RASSF5 modulates the Hippo pathway by phosphorylating its component pro-apoptotic mammalian STE20-like protein kinase 1/2 (MST1/2), leading to the activation of large tumor suppressor kinase 1/2 (LATS1/2) and the protein degradation of YAP1, controlling the expression of genes involved in proliferation (e.g., proliferating cell nuclear antigen; PCNA), invasion (e.g., matrix metallopeptidase 9; MMP9), and apoptosis (e.g., p53) [29]. Besides MST1/2-mediated RASSF5 functions on cell proliferation and apoptosis, RASSF5 was also reported to function independently of ras or MST1/2 in lung cancer cells [30]. RASSF5 was also shown to be involved in apoptosis, mediated by the tumor necrosis factor α (TNF-α), TNF-related apoptosis-inducing ligand (TRAIL), and CD40 ligand [31,32]. Overall, RASSF5 may exert tumor-suppressive functions through multiple context-dependent mechanisms. In ovarian cancer cells, *RASSF5* downregulation was shown due to CpG hypermethylation in the *RASSF5* promoter. In contrast, *RASSF5* upregulation by ten-eleven translocation 1 (TET1)-mediated demethylation may result in the suppression of ovarian cancer cell proliferation [33].

ADORA1 is a G-protein coupled receptor (GPCR) for adenosine, typically mediated by the G-proteins Gi and Go [34]. The role of ADORA1 in cancer remains controversial. ADORA1 was shown to suppress proliferation and induce apoptosis in CW2 colon cancer and MCF-7 breast cancer cells [34]. Conversely, ADORA1 was reported to promote the growth and survival of 786-O and ACHN kidney cancer cells, or MDA-MB-468 breast cancer cells [34,35]. Notably, treatment with the ADORA1 antagonist SLV320 partially rescued the adenosine-mediated decrease in A2780 cell survival, suggesting that ADORA1 may play a tumor-suppressive role in ovarian cancer [36].

A question remains how *OIN1* modulates the expression of *RASSF5* and *ADORA1* in ovarian cancer. Previous literature showed that *LUCAT1* and *TUG1* are examples of oncogenic lncRNAs that suppress apoptosis in ovarian cancer: the former promotes the proliferation and migration of cancer cells through the *miR-612*/homeobox A13 (HOXA13) axis [37], and the latter decreases apoptosis by targeting *miR-29b-3p*, leading to paclitaxel resistance [38]. Similarly, *OIN1* may also have a possibility to promote ovarian cancer progression by targeting particular miRNAs. In terms of sequence similarity, we did not observe substantial homology between *OIN1* and *RASSF5* or *ADORA1* mRNAs, suggesting that *OIN1* may interact with *RASSF5* and *ADORA1* mRNAs through a mechanism other than direct RNA-RNA binding. Recent study of long terminal repeat (LTR) retrotransposon-derived lncRNA *p53-regulated lncRNA for homologous recombination repair 1* (*PRLH1*) in p53-mutated hepatocellular carcinoma showed that the lncRNA functions as a homologous recombination-promoting factor [39]. Interestingly, we found that *OIN1* may have a similarity with human endogenous retrovirus (HERV)-like long terminal repeat (LTR) retrotransposon ERV1 clade [40] as shown by the UCSC Genome Browser. The oncogenic role of ERV1 has been shown as high ERV1 expression, and in kidney cancer was shown to be associated with worse patient survival outcomes [40]. Given that *OIN1* was originated from an endogenous retrovirus-like sequence, future study may reveal whether the lincRNA may affect the homologous recombination or immune reactions contributing to the pathophysiology of ovarian cancer.

In the present study, we examined the role of *OIN1* in ovarian cancer mainly using A2780 and SKOV3 cells. Considering the heterogeneity of ovarian cancer, studies with other ovarian cancer cells may also provide useful information to understand the oncogenic role of *OIN1* (e.g., *OIN1* overexpression in ovarian cancer cells with low expression of *OIN1*, such as OVCAR3 and ES2). Analyzing the role of *OIN1* in xenograft models using ovarian cancer cells other than A2780 will provide useful information to characterize the in vivo role of *OIN1*. Recently, three-dimensional cultures of patient-derived cancer cells (PDCs) and patient-derived xenograft (PDX) models have been applied to preclinical studies because they usually retain the properties of original tumors [41,42,43,44,45]. Characterizing *OIN1* in in vitro experiments with PDCs and their xenograft models or PDX models will be useful to elucidate the precise role of this lincRNA. For clinical applications of *OIN1* to ovarian cancer treatment, it will be important to examine whether intravenous siOIN1 injections substantially repress tumor formation of ovarian cancer models. As we recently demonstrated that intravenous injections of an siRNA targeting an oncogenic lncRNA *thymopoietin antisense transcript 1* (*TMPO-AS1*) efficiently suppressed the xenograft tumor growth and lung metastasis derived from breast cancer cells [46], the therapeutic efficacy of siOIN1 can be evaluated in similar ovarian cancer xenograft models or patient-derived cancer models.

## 4. Materials and Methods

### 4.1. RNA-Seq Analysis of Clinical Specimens from Normal and Ovarian Cancer Tissues

Study protocols and patient consent were approved by the Institutional Review Board of Saitama Medical University International Medical Center (#12-096, #13-165). Ovarian cancer and normal tissue specimens were obtained from patients who underwent surgery for primary ovarian tumor. Detailed information for the clinical specimens and RNA-seq methods were described previously [15,16]. Expression values of RNA-seq were quantified as RPKM based on RefSeq and NONCODE (http://www.noncode.org/ [accessed on 22 January 2021]) gene models [18]. Among transcripts mapped to NONCODE v4 gene sets, we selected 10 putative differentially expressed lincRNAs, which were particularly upregulated in ovarian cancer compared with normal tissues by ≥10-folds at an FDR adjusted *p*-value, or *q*-value threshold <0.05 (Table 1). Among the 10 lincRNAs, *NONHSAT013448* designated as *OIN1* in this study exhibited the highest expression in ovarian cancer tissues. We screened positively or negatively *OIN1*-correlated Refseq genes showing correlation coefficients (*r*) between *OIN1* as ≥0.6 or ≤−0.45. Biological pathways enriched among the *OIN1*-associated genes were analyzed using DAVID Bioinformatics Resources 6.8 (https://david.ncifcrf.gov/summary.jsp [accessed on 22 January 2021]) [24].

### 4.2. Human Ovarian Cancer Cell Culture

OV90, OVCAR3, and SKOV3 ovarian cancer cells, A2780 ovarian cancer cells, and ES2 ovarian cancer cells were grown in Dulbecco’s modified Eagle’s medium (DMEM), in RPMI 1640 and DMEM/F12, respectively, supplemented with 10% fetal bovine serum (FBS), 100 U/mL penicillin, and 100 μg/mL streptomycin at 37 °C in a 5% CO_2_ atmosphere. All cell lines were authenticated using short-tandem-repeat (STR) analysis (BEX, Tokyo, Japan).

### 4.3. siRNA and Plasmid Transfection

The custom small interfering RNAs (siRNAs) against *OIN1* synthesized by Sigma-Aldrich (St Louis, MO, USA) were as follows: siOIN1 #1, 5’-GCUCAGCUCACGGCUUCUACC-3’ (sense) and 5’-UAGAAGCCGUGAGCUGAGCUC-3’ (antisense); siOIN1 #2, 5’-GACAGGAGACUCCAGAAAAGG-3’ (sense) and 5’-UUUUCUGGAGUCUCCUGUCUG-3’ (antisense). The control siRNA (siControl) was synthesized at RNAi Inc. (Tokyo, Japan) [47]. Cells were transfected with siRNAs (10 nM) using Lipofectamine RNAiMax (Thermo Fisher Scientific, Waltham, MA, USA) or transfected with the indicated plasmids using Lipofectamine 3000 reagent (Thermo Fisher Scientific) or FuGene HD transfection reagent (Promega, Madison, WI, USA). After 24, 48 or 72 h incubation, cells were harvested for qRT-PCR.

### 4.4. RNA Extraction and qRT-PCR

Total RNA was extracted from the ovarian cancer cells and A2780-derived xenografted tumors using ISOGEN reagent (Nippon Gene Co., Toyama, Japan) or Sepasol-RNA I Super G (Nacalai Tesque, Kyoto, Japan). One microgram of total RNA was reverse-transcribed to single-stranded cDNAs using SuperScript III (Thermo Fisher Scientific) or PrimeScript™ RT reagent kit (perfect real time) (Takara Bio Inc., Shiga, Japan). qRT-PCR was carried out on a StepOnePlus™ real-time PCR System (Thermo Fisher Scientific) using KAPA SYBR FAST qPCR Kit (KAPA Biosystems, Wilmington, MA, USA) with gene-specific primers. Relative RNA levels were analyzed using the ΔΔCt method according to the manufacturer’s protocol and normalized to *GAPDH*. Primers used for qPCR are listed in Appendix A.

### 4.5. Cell Proliferation Assay (DNA Assay)

A2780 and SKOV3 cells were plated at 3000 or 1000 cells/well, respectively, in 96-well plates. After 24 h, these cells were transfected with the indicated siRNAs or plasmids as described above. Cells were collected at the indicated days after cell plating. To evaluate cell proliferation ability, the extracted DNA was stained with Hoechst 33258 pentahydrate (Thermo Fisher Scientific) at a final concentration of 5 μg/mL. DNA content in each well was measured using a 2030 ARVO X5 multilabel plate reader or VICTOR Nivo multimode microplate reader (Perkin Elmer, Foster City, CA, USA) [48].

### 4.6. Apoptosis Assay

A2780 and SKOV3 cells were plated at 3 × 10^5^ or 1 × 10^5^ cells/well, respectively, in 6-well plates. After 24 h, the cells were transfected with the indicated siRNAs at a final concentration of 10 nM and collected 72 h after transfection. Apoptotic cells were stained with a FITC Annexin V Apoptosis Detection Kit I (BD Biosciences, San Jose, CA, USA) following the manufacturer’s instructions. Annexin V- and propidium iodide (PI)-positive cells were analyzed with BD FACSCalibur (BD Biosciences) [48].

### 4.7. Western Blotting

Whole cell lysates were prepared from A2780 or SKOV3 cells transfected with indicated siRNAs for 48 or 72 h. Western blotting using anti-cleaved PARP1 (Abcam, Cambridge, UK) and β-actin (Sigma–Aldrich) antibodies and subsequent detection were performed as described [46].

### 4.8. In Vivo Tumor Formation and siRNA Treatment

All animal experiments were approved by the Animal Care and Use Committee of Saitama Medical University, and performed following the institutional Guidelines and Regulations. Female athymic mice (BALB/cAJcl-*nu*/*nu*) were purchased from CREA Japan Inc (Tokyo, Japan). A2780 cells (5 million cells per mouse) were mixed with an equal volume of Matrigel matrix (Corning, Corning, NY, USA) and injected subcutaneously into the side flank of 10-week-old female athymic mice. We inoculated A2780 cells subcutaneously into 20 female athymic mice, and tumors were generated in 15 mice. Then, we randomly assigned these 15 mice to the siControl (*n* = 7) or siOIN1 #1 groups (*n* = 8). siControl or siOIN1 #1 (5 μg each) was prepared with the GeneSilencer reagent (Gene Therapy System, San Diego, CA, USA) as described previously [49] and injected into the generated tumors twice a week. Three dimensions of tumor were measured twice a week, and tumor volumes were calculated using the following formula: 0.5 × largest dimension × intermediate dimension × shortest dimension.

### 4.9. Statistical Analysis

Statistical analysis was performed using two-way analysis of variance (ANOVA), or unpaired two-tailed Student’s *t*-test, as indicated. JMP 9.0.0 (SAS Institute) was used for statistical analysis.

## 5. Conclusions

In summary, we showed that *OIN1* is a functional lincRNA abundantly expressed in ovarian cancer and functions as a tumor-promoting molecule by suppressing apoptosis, suggesting that *OIN1* may be served as a potential diagnostic and therapeutic target for ovarian cancer.

## Figures and Tables

**Figure 1 ijms-22-11242-f001:**
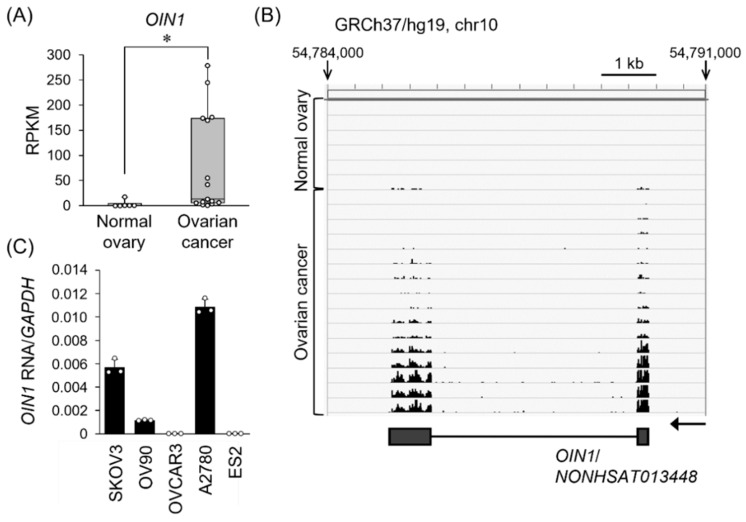
Overexpression of lincRNA *OIN1* in ovarian cancer. (**A**) *OIN1* RNA (registered as *NONHSAT013448* in NONCODE) is overexpressed in ovarian cancer. RPKM (reads per kilobase of transcript length per million mapped reads) values of *OIN1* RNA were estimated by RNA-seq analysis using ovarian cancer (*n* = 15) and normal tissues (*n* = 6) [15,16]. Data are presented by box plots. *, *q* < 0.01. (**B**) The mapping data of RNA-seq reads derived from normal ovary tissues and ovarian cancer on the *OIN1* locus. A genome browser snapshot of the *OIN1* region (chr10:54,784,000–54,791,000 in hg19) was shown with the schematic representation of *OIN1* gene. The grey boxes indicated the exons of *OIN1*. The arrow shows the direction of *OIN1* gene. (**C**) *OIN1* RNA expression levels in ovarian cancer cell lines. qRT-PCR was performed to quantify the expression levels of *OIN1* RNA normalized to *GAPDH* mRNA levels. Data are presented as mean ± SD (*n* = 3).

**Figure 2 ijms-22-11242-f002:**
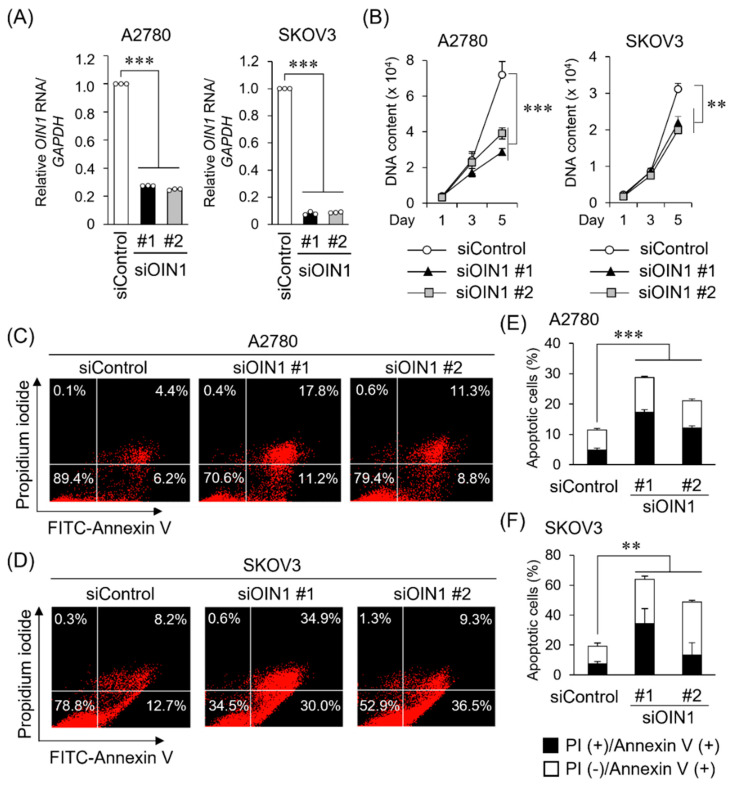
*OIN1* regulates proliferation and apoptosis in ovarian cancer cells. (**A**) Knockdown efficiencies of siRNAs targeting *OIN1* (siOIN1 #1 and #2) in A2780 and SKOV3 cells 48 h after siRNA transfection were analyzed by qRT-PCR. Relative *OIN1* RNA expression levels were normalized to *GAPDH* mRNA levels and presented as mean fold change ± SD compared with control siRNA (siControl) in each cell type (*n* = 3). (**B**) Inhibitory effects of *OIN1* knockdown in A2780 and SKOV3 cell proliferation were analyzed by DNA assay. Data are presented as mean ± SD (A2780, *n* = 5; SKOV3, *n* = 3). (**C**–**F**) Promoting effects of *OIN1* knockdown in apoptosis of A2780 (**C**) and SKOV3 (**D**) cells 72 h after siRNA transfection were analyzed by flow cytometry with propidium iodide and annexin V. Percentages of annexin V-positive populations of A2780 (**E**) and SKOV3 (**F**) cells treated with indicated siRNAs were quantified (*n* = 3). **, *p* < 0.001; ***, *p* < 0.0001, two-way ANOVA.

**Figure 3 ijms-22-11242-f003:**
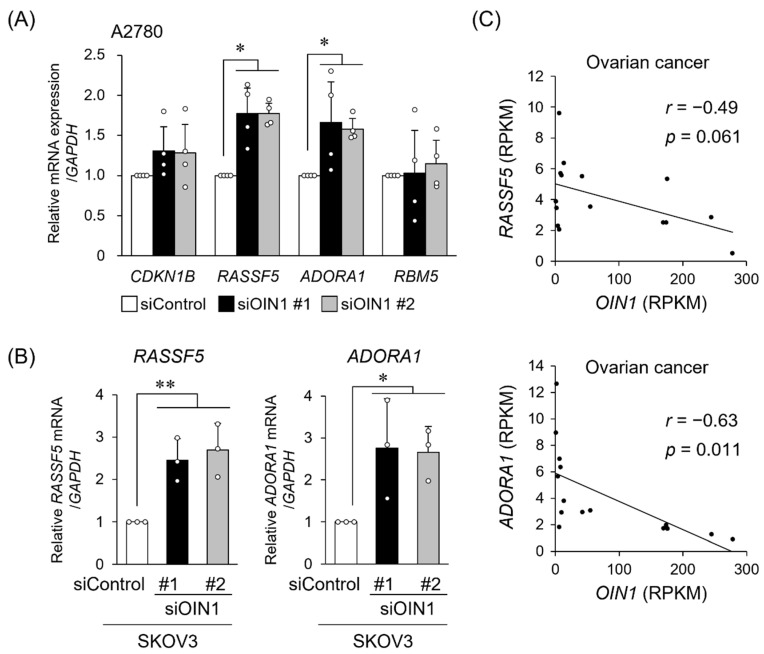
*OIN1* regulates expression levels of *RASSF5* and *ADORA1.* (**A**) Increased expression of *RASSF5* and *ADORA1* in *OIN1*-downregulated A2780 cells. A2780 cells were transfected with *OIN1* siRNAs (siOIN1 #1 and #2) or siControl. Relative expression levels of selected genes 48 h after siRNA transfection were quantified by qRT-PCR and normalized to *GAPDH* mRNA levels. Data are presented as mean fold change ± SD in each gene group (*n* = 4). (**B**) Increased expression of *RASSF5* and *ADORA1* in *OIN1*-downregulated SKOV3 cells. SKOV3 cells were transfected as above, and expression levels of *RASSF5* and *ADORA1* mRNAs 72 h after siRNA transfection were quantified by qRT-PCR. Data are normalized to *GAPDH* mRNA levels and presented as mean fold change ± SD in each gene group (*n* = 3). *, *p* < 0.05; **, *p* < 0.01; two-way ANOVA. (**C**) Relationship between expression levels of *RASSF5* or *ADORA1* mRNA and *OIN1* RNA analyzed using our RNA-seq data for clinical ovarian cancer specimens (*n* = 15) [15,16].

**Figure 4 ijms-22-11242-f004:**
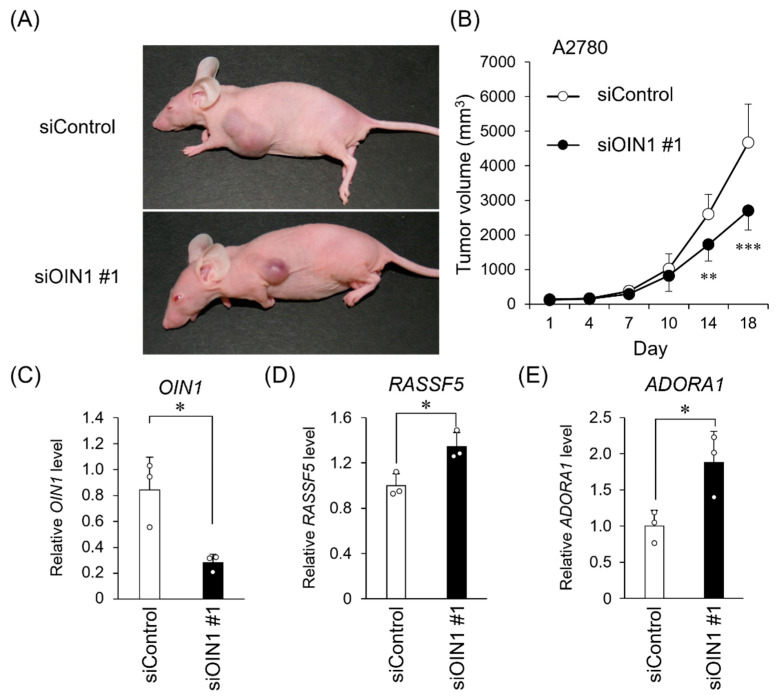
*OIN1* siRNA inhibits tumor formation of ovarian cancer xenografts. (**A**) Athymic mice were subcutaneously xenografted with A2780 cells and, then, siOIN1 #1 (*n* = 8) or siControl (*n* = 7) was injected into the tumors twice a week. Representative images of generated tumors after 18 days are shown. (**B**) Tumor volumes are presented as mean ± SD. (**C**–**E**) Relative expression levels of *OIN1* RNA (**C**), *RASSF5* (**D**), and *ADORA1* (**E**) mRNAs in the tumors dissected 18 days after the start of siRNA injection were quantified by qRT-PCR and normalized to *GAPDH* mRNA levels. Data are presented as mean fold change ± SD versus siControl in tumors (siControl, *n* = 3; siOIN1 #1, *n* = 3). *, *p* < 0.05; **, *p* < 0.01; ***, *p* < 0.001; Student’s *t*-test.

**Table 1 ijms-22-11242-t001:** Differentially upregulated lincRNAs in ovarian cancer versus normal ovary tissues ^a^.

NONCODE ID	Chr	Start	End	Strand	Alias	Normal Ovary		Ovarian Cancer		Fold Change ^b^	*q*-Value
						Mean RPKM	SEM RPKM	Mean RPKM	SEM RPKM		
*NONHSAT013448*	10	54,785,023	54,789,855	-	** *OIN1* **	3.0	2.6	79.2	24.8	26.1	5.6 × 10^−3^
*NONHSAT099419*	4	182,443,812	182,444,154	+		3.2	2.7	40.4	5.5	12.7	2.9 × 10^−4^
*NONHSAT017219*	11	287,304	288,298	+		0.6	0.2	38.6	18.3	62.9	1.0 × 10^−2^
*NONHSAT027397*	12	26,383,751	26,472,653	-		0.5	0.2	27.4	11.7	51.5	1.6 × 10^−3^
*NONHSAT015316*	10	85,926,985	85,931,832	-	*CERNA2/HOST2*	0.8	0.7	17.2	5.9	20.2	7.4 × 10^−4^
*NONHSAT032437*	13	23,477,401	23,493,348	+		0.3	0.1	9.2	3.0	31.0	1.5 × 10^−4^
*NONHSAT080725*	20	60,880,487	60,881,452	-		0.5	0.2	8.8	3.8	19.2	1.3 × 10^−3^
*NONHSAT122583*	7	104,581,509	104,602,507	+		0.3	0.2	8.2	3.7	27.0	3.0 × 10^−2^
*NONHSAT061517*	19	15,939,789	15,947,064	+	*UCA1*	0.6	0.5	6.7	2.0	11.3	8.1 × 10^−3^
*NONHSAT018088*	11	13,002,033	13,005,839	-	*LINC00958*	0.6	0.5	5.7	0.9	10.3	2.1 × 10^−2^

^a^ Differentially upregulated lincRNAs in ovarian cancer were selected by the criterion described below: among transcripts mapped to NONCODE v4 gene sets, we selected 10 putative differentially expressed lincRNAs, which were particularly upregulated in ovarian cancer compared with normal tissues by ≥10-folds at an FDR *q*-value threshold <0.05. ^b^ Fold change of RPKM values in ovarian cancer versus normal ovary tissues. RPKM: reads per kilobase of transcript length per million mapped reads.

## Data Availability

The data used to support the findings of the present study are available from the corresponding author upon request.

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
