# Peer review of "Long Intergenic Noncoding RNA OIN1 Promotes Ovarian Cancer Growth by Modulating Apoptosis-Related Gene Expression"

_ijms, 2021, doi:10.3390/ijms222011242_

Round 1

Reviewer 1 Report

ijms-1409600 revealed the role of long intergenic noncoding RNA OIN1 in ovarian cancer growth. The authors suggested that OIN1 is highly expressed in ovarian cancer tissues. Moreover, the authors verified that inhibition of OIN1 suppressed proliferation and promoted apoptosis in ovarian cancer cells. The effects of OIN1 were also confirmed in a xenograft model. The topic is interesting and the experimental design is well organized. The manuscript is also well written, but I recommend the presentation of some experimental data to reinforce the conclusions of this study.

  1. I recommend the authors analyze the expression of apoptosis-related proteins (e.g. PARP, Bcl-2, Bax, caspases, etc.) in response to OIN1 knockdown in ovarian cancer cells.
  2. Can the authors present tumor images collected from the xenograft model? A listing of images of tumors removed from mice would more intuitively show the effect of siOIN1.

Reviewer 2 Report

Takeiwa et al. through RNAseq analysis identified a novel LincRNA, named OIN1, which is over-expressed in Ovarian cancer (OC) tissues compared with the normal ones. Further, they showed that KD of OIN1 enhanced apoptosis and decreased proliferation of OC cell lines. The authors, using RNAseq, and in silico analysis, identified which apoptosis-related genes were inversed correlated with the expression of OIN1 and validated their results in OC cell lines. Lastly, they performed in vivo studies by injecting A2780 cell line and showed that after intratumoral injection of siROIN1 the tumor volume was decreased most probably due to an increase of RASSF5 and ADORA1 apoptosis/proliferation-related genes. 

Overall, the work is interesting and well performed and gives some novel insights of lincRNA OIN1 involvement in the progression of OC, which could become a new therapeutic target in OC patients.

Nevertheless, I have some major and minor points regarding the present study:

Major points:

1. Line 129, Why the authors over-expressed OIN1 in A2780 and SKOV3 cell lines which already express high levels of this lincRNA? Did they perform over-expression of OIN1 in OVCAR3 or ES2 in cell lines, in which OIN1 is absent? How the authors explain this?

2. Section 4. 2: The authors should describe the type of OC cell lines and if these cell lines are characterized by TP53/BRCA mutations. E.g. Is TP53 mutated in SKOV3 and A2780 cell lines? How does this correlate with the high expression of OIN1 in these two cell lines and the same thing for the other three cell lines used in this study?

3. In the figure legends, the time points of harvested cells for each experiment are missing. Please, specify this in all figure legends.

4. Why did the authors use only A2780 for xenografts? At which time point after intratumor siRNA injection the authors noticed the increase of RASSF5 and ADORA1? Does this time point correspond to the in vitro experiment in the same cell line, showed in Figure 3A?

5. Did the authors used other siRNA concentrations for intratumor injections? And why the authors have chosen intratumor over intravenous injections? Thinking of a siRNA-based therapy in OC patients, IV injections are more likely to be used in clinic.

6. Line 314: from the graph of tumor volume growth over time results that the tumor volumes were measured every three days and not once a week.  Is it 3 or 7 days? 

Minor points:

1. The authors should explain NONHSATO13446 in the figure legend and what the grey boxes are in Figure 1 B.

2. Why there were only 7 and not 8 mice in the siRcontrol group? Also, specify which were the three dimensions at line 314.

3. In the introduction at line 37, the authors can cite a couple of very recently published papers regarding lncRNAs and miRNAs involved in OC progression that can be used as biomarkers. PMID: 34367239 and PMID: 34573382

Round 2

Reviewer 1 Report

The authors improved the manuscript by considering all comments.

Reviewer 2 Report

The authors have replied in an excellent and exhaustive way my comments.